# Vowel Quality in Xiang Non-Lexical Hesitation Markers: New Forms of Typological Evidence?

Robert Marcelo Sevilla

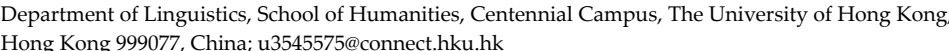

Department of Linguistics, School of Humanities, Centennial Campus, The University of Hong Kong, Hong Kong 999077, China; u3545575@connect.hku.hk

**Abstract:** Xiang (hsn) remains a poorly understood grouping within Sinitic, with no satisfactory conclusions on how to demarcate its boundaries or define its subgroupings. One general observation is that there is a rough typological split between the Northeast and Southwest related to contact from northern- and southern-type Sinitic varieties, respectively, which can be supported with phonological, lexical, and syntactic evidence. It is predicted here that an additional source of evidence can be found in the phonetic features of hesitation markers (HMs; 'fillers', 'speech disfluencies', etc.), which tend towards the central area of the vowel space (approaching [ə], [ɤ], [e], etc.) but still conform to the phonologies of the languages in which they occur. This study introduces a novel three-way division of Xiang in terms of phonemic central vowels found in open syllables (either [ə/ɤ], [e/ɛ], or both) which is then evaluated against the vocalic quality found in HMs to determine whether they can be used as evidence for Xiang internal typology. Data are gathered from 47 speakers representing 16 Xiang localities, distributed across Hunan province, recorded performing the Pear Stories paradigm, with 304 hesitation markers extracted. Features reported on include vowel quality (F1-F2), tonal contour (F0), and duration (ms). It is found that Xiang HMs demonstrate four distinct vowel qualities, but that their distribution does not neatly fit established taxonomic schemes; however, the evidence does support the transitional status of Xiang varieties as a site of mixture of northern and southern Sinitic features.

**Keywords:** hesitation markers; Xiang; phonetics-phonology; typology





## 1. Introduction

The use of hesitation markers (HMs; fillers, filled pauses, disfluencies, etc.) as evidence of crosslinguistic typological trends has been explored in the past (Candea et al. 2005; Vasilescu et al. 2005; Wieling et al. 2016; Dingemanse 2017; Dingemanse and Woensdregt 2020), but few if any studies have been conducted on whether or not they can inform existing typological[1] classificatory models. Particularly within Sinitic, HMs have received little attention, with the exception of a few key studies (Zhao and Jurafsky 2005; Yuan et al. 2016) all dealing with Mandarin. Meanwhile, within Sinitic the Xiang group has often not been provided the same amount of attention as other, better-known varieties such as Yue or Min. The general consensus seems to be to divide Xiang into northeastern and southwestern areas (Yuan 1960; Wu 2005), but the features these two areas share to the exclusion of others is rather small. In order to further understanding on the typological subgrouping of Xiang, as well as contribute to understanding on the vocalic quality of hesitation markers, the present study seeks to explore the plausibility of using the vowel quality of hesitation markers as evidence for the internal typological taxonomy of Xiang.

Xiang has been the focus of much work in Sinitic dialectology, with many studies being dedicated to the precise classification of the grouping, both within Sinitic generally and in terms of its constituent subgroups (Yuan 1960; Language Atlas of China 1988, 2012; Zhou and You 1988; Chen and Bao 2007; Coblin 2011; Bao 2017; among others). A lot of this work (Coblin 2011 being a key exception) toes the line between historico-comparative

classification, used to determine closer genetic affiliations, and typological classification, which aims to define groups based on bundles of shared features, and it can be often unclear on which side a particular study lies. For instance, Yuan (1960) originally defined the Southwestern Xiang subgroup ('Old' Xiang in his terminology) as any variety of Chinese which retained the Middle Chinese voiced plosives in all tones. However, as preservation of voiced plosives is a retention rather than an innovation, it provides insufficient grounds upon which to motivate a genetic classification. In addition, Xiang as a whole and many of the subgroups defined for Xiang lack a substantial list of uniquely shared features, whether retentions or innovations. One of the factors contributing to this lack of shared features is the 'transitional' (Wu 1999; Chappell 2001; Szeto and Yurayong 2021) status of Xiang, where southern and northern Sinitic features are mixed. Across all of these studies (again, with Coblin 2011 as the exception), one general consensus is apparent: Xiang can be divided into two large typological[2] areas, a northeastern, innovative one, and a southwestern, conservative one. Modifications on this schema have added additional groupings, but the number of shared features remains rather small (see Chen and Bao 2007, 2012).

Hesitation markers are still often not thought of as 'words' in the usual sense, often labelled 'disfluencies' (Shriberg 2001; McDougall and Duckworth 2017) or 'filled pauses' (Gick et al. 2004) given their primary function as placeholders while one formulates the rest of an utterance. Two broad types of HMs are distinguished in the present piece: one having a lexical source and therefore a dual function, such as the English 'like' or 'so' (hence, 'lexical HM'), and another having no such source, such as the English 'uh' or 'um' (hence, 'non-lexical HM'). While few would argue that the former type do not consist of 'words', the latter has a somewhat more ambiguous status. The high incidence of non-lexical HMs in natural speech and their conventionalized form in different languages (Clark and Fox Tree 2002) has led to a renewed appraisal of their status as words, specifically as discourse markers (following definitions in Maschler and Schiffrin 2015). However, unlike lexical HMs, it is also clear that non-lexical HMs have a special status which differentiates them from regular vocabulary; this is evidenced by how remarkably similar non-lexical HMs are across languages (Candea et al. 2005; Dingemanse and Woensdregt 2020), tending to cluster towards vowels in the mid-to-low acoustic space. Meanwhile, the language-specific quality non-lexical HMs[3] have, as evidenced by a better-than-chance ability to identify languages by HM acoustic quality (Vasilescu et al. 2005), are also predicted to represent viable ground for typological research. This is particularly true for Xiang, given that it is precisely in the mid-vowels that divisions can be drawn in terms of which vowels are contrastive in open syllables.

Given these facts, the following questions are raised: what segmental/suprasegmental patterns can be observed for HMs in the transitional Xiang grouping, which represents a typological link between the Northern and Southern Sinitic languages? Will vowel quality demonstrate northern characteristics, southern characteristics, or reflect some third paradigm? Can HM vowel quality serve as an additional piece of evidence for the internal typological affiliation of a particular dialect?

Through analysis of a corpus of 47 narratives from 16 Xiang-speaking localities, this study aims to explore the vocalic quality of hesitation markers as they vary across the grouping. A novel, three-way division of Xiang dialects based on mid-vowel contrasts in open syllables is introduced. In so doing, the present research seeks to explore whether sub-lexical vocabulary like hesitation markers can be used to expand the number of features used in defining typological splits among dialects in Xiang. Findings suggest that while division into discrete, categorical groupings is unlikely given the variability of HMs and other confounding factors, more gradient divisions can be established. In addition, the present research has implications for our understanding of how hesitation markers manifest in naturalistic speech, including an exploration of how HM acoustics succeed or fail to reveal more general phonological patterns.

## 2. Background

Sinitic can be divided into two large typological areas: a northern area and a southern area (Yue-Hashimoto 1976; Norman 1988; Chappell 2001, 2015; Szeto 2019; Szeto and Yurayong 2021), representatives of which can be found in Mandarin and Cantonese, respectively.[4] Following Norman (1988), the northern area includes varieties of Mandarin, and the southern area includes varieties of Yue, Hakka, and Min. These regions can be defined at many levels of linguistic description, including phonology (complexity of the tonal system, treatment of Middle Chinese voiced plosives, loss of coda consonants, etc.), syntax-morphology (complexity of the classifier system, type of comparative construction, modifier-head ordering, etc.), and lexicon (choice and variety of negation particles, degree of monosyllabicity, origin of the comparative marker, etc.). In between these two areal regions lie the remaining three traditional groupings of Sinitic: Wu, Gan, and Xiang. This central region is thought to share features of both the north and the south, which contributes to the 'transitional' characterization of the Sinitic varieties spoken in these regions (Wu 2005; Chappell 2015; Szeto and Yurayong 2021).

While this areal typology has traditionally been defined in terms of well-established grammatical, phonological, and lexical features, the question this study seeks to answer is whether our understanding of this typological scheme and the notion of a 'transitional' Xiang can be enriched through exploration of acoustic features associated with hesitation markers, in particular the feature of hesitation marker vowel quality. HMs are small, discourse-moderating particles that signal a pause in speech, often composed of a mid-central vowel or nasal consonant with low pitch (Candea et al. 2005; Zhao and Jurafsky 2005; Yuan et al. 2016; Dingemanse and Woensdregt 2020), and the exact quality of the vowel in question is usually determined by language-specific phonological patterns, such as which vowels a language allows in open syllables. Returning to Sinitic, in open syllables Mandarin and other northern varieties restrict central vowels to mid-back [ɤ] (or mid-central [ə]), while Cantonese restricts them to mid-front [ɛ] and its rounded counterpart (Shi et al. 2015),[5] but it is unclear how these vowels pattern in Xiang, and, by extension, how they manifest in HMs (the typology of HMs will be further discussed in Section 2.3). The exact quality seen for these items in Xiang can potentially tell us something about the relative clustering of northern versus southern features in a particular dialect.

Central Sinitic languages are predicted to lie somewhere between the north and south in terms of their typological features; in particular, Xiang varieties have vocalic systems which display features of both. For instance, the Chengbu dialect has an open syllable contrast between mid-central and mid-front vowels (Bao 2017, pp. 152–174), the Yiyang dialect has only a mid-central vowel (Cui 1998), and Xiangtan (Wang 2013) has only a mid-front vowel. This mix of features raises questions as to where the form of HMs cluster in Xiang and whether they can in some way illuminate its internal typological subgrouping.

### 2.1. Classificatory Overview of Xiang

The area where 'Xiang' is spoken covers most of central Hunan province and parts of northern Guangxi, primarily following the Xiang River, which gives the grouping its name. The most common linguistic definition refers to those dialects which have uniform treatment of the Middle Chinese voiced plosives as either voiced or voiceless unaspirated (Yuan 1960; Norman 1988; Ho 2015). It is surrounded on most sides (to the north, west, and south) by varieties of Southwestern Mandarin, as well as varieties of Gan and Kejia in the east. There is also a small, poorly researched 'Waxiang' area in the west (Wu 2005, p. 375). Much work has focused on better defining the group and its subgroups (Yuan 1960; Bao 1985; Language Atlas of China 1988, 2012; Zhou and You 1988; Wu 2005; Chen and Bao 2007; Coblin 2011). It is important to stress that what will be covered in what follows are not genetic, but rather typological, conceptualizations in the strict sense (although some authors may not agree with this assessment); as work by Zhou and You (1988) and Coblin (2011) show, the genetic tenability of the Xiang family is suspect, and the term 'Xiang' here is thus used as an areal or typological convention.

Yuan (1960) originally divided the grouping into two main dialect areas, roughly corresponding to a north-south split: a 'New' and an 'Old' group, based on loss or retention of the Middle Chinese voiced obstruents, respectively. This simple taxonomy is complicated by the fact that Old dialects have varying degrees of voicing preserved; for instance, some may have a fully voiced three way contrast between /b d g/, others only /b d/, while others only have /d/ (Norman 1988). Another complication is the fact that voiced plosive preservation is not unique to Xiang, as it is found in Wu dialects as well (Kurpaska 2010; Ho 2015). This original taxonomy is expanded on by the first edition of the Language Atlas of China (1988), which defines Xiang as those varieties which have unaspirated reflexes of the Middle Chinese voiced obstruents in all tones by adding the 'Jixu' grouping (made up of western Xiang dialects originally of the Old Xiang grouping) and by renaming the Old and New families 'Loushao' and 'Changyi', respectively.[6] The definitional criteria for these three groupings are: Changyi has unaspirated voiceless reflexes of the MC voiced obstruents, Loushao has fully voiced reflexes, and Jixu has voiced reflexes in syllables in the level tone category, but voiceless ones in the oblique tone category. Another change includes the incorporation of Anhua into the Loushao group, despite its lack of voiced plosive initials in all its varieties (Zhou 2016). In addition, the atlas notes that there are exceptions to the loss of voiced plosives in Changyi, namely Hengdong and Hengshan (more on these problematic cases later).

Other classificatory models use a wider variety of criteria to determine groupings, of which Chen and Bao (2007) is a good example (repeated in Bao 2017, pp. 22–36). This scheme includes five groupings: Changyi (northeast), Loushao (southwest), Hengzhou (central), Chenxu (west), and Yongquan (south), which is summarized in Figure 1 ("Dialects of Hunan per Chen and Bao (2007)" by Alan Tsukuba under: https://creativecommons.org/licenses/by-sa/4.0/deed.en, accessed 21 November 2021). Bao (2017, pp. 22–36) lists the phonological criteria used for the classification, which once again include voiced plosives and tonal contrasts as a major feature, but with the novel addition of vocabulary as a diagnostic feature. For instance, features treated as unique for Hengyang (a subgroup within Hengzhou) include (unsurprisingly) the voiceless unaspirated reflexes of Middle Chinese voiced obstruents; a series of six lexical tones with pitch values similar to Shaoyang; and unique lexical items for the third person pronoun [tɕi33], 'son' [lai33 tɕi33], and 'what' [ma33 ko22]. What is particularly interesting is the choice to group Hengyang, Hengdong, and Hengshan into their own 'Hengzhou' grouping, as well as Yongzhou into its own 'Yongquan' grouping, due to their perceived transitional status between the better-known Loushao and Changyi groupings.

The updated second edition of the Language Atlas of China (2012, p. 19) also adopts this five-way classificatory scheme. It appears this was done in response to Chen and Bao's (2007) findings, as the updated version of the Atlas also includes an article detailing their criteria (Chen and Bao 2012). Therefore, Chen and Bao's approach represents the most up-to-date understanding of the internal classification of Xiang, as far as the author is aware.

What is clear about Xiang, through all of these revisions, is that there appears to be some typological north(east)-south(west) divide (typified by Changsha and Shaoyang, respectively), with certain central transitional areas which breed complications (such as Hengyang and Yongzhou). Hengyang (and environs) is divergent to the point of requiring its own grouping (Chen and Bao 2007, 2012). This is analogous to the typological situation in the Sinitic family as a whole, with well-established (or comparatively better-established) northern and southern groupings (typified by Mandarin and Yue, respectively) and with certain central varieties complicating matters (such as Xiang). This general trend has also been found in more recent typological research such as that of Szeto and Yurayong (2021), who find a general north-south divide in Xiang as well.

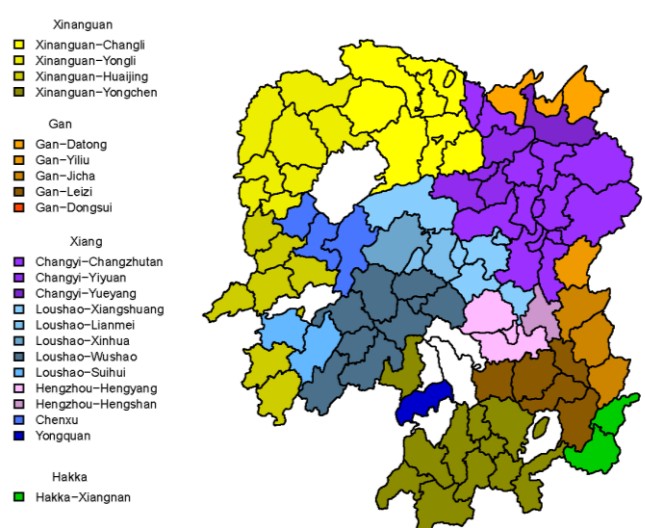

**Figure 1.** Classification of Xiang varieties (shades of blue = Loushao, Chenxu, Yongquan; purple = Changyi; pink = Hengzhou), following Chen and Bao (2007).

*2.2. Comparison of Vowel Inventories*

One overarching typological feature in Sinitic which has not received much attention is the distribution of unrounded mid-vowels in open CV syllables. We know, for instance, that in Putonghua (PTH; Standard Mandarin) only mid-back [ɤ] occurs in open syllables without an accompanying glide (Duanmu 2007, p. 37), and that Cantonese (Matthews and Yip 2013) allows only mid-front lax [ɛ], but the trend is more widespread.

Lee and Zee (2014) mention that in their survey of 70 Sinitic varieties, 52 had vocalic systems with either [e] or [ɛ] as the only contrastive unrounded mid-vowel in open CV syllables, while 14 had [ɤ] and 12 had [ə] (although presumably not both). If one goes over Sinitic varieties by their relative northern or southern affiliation following Norman (1988, pp. 181–3), we can observe a general north-south divide. In the UPSID database (Maddieson and Precoda 2018), one can see that Min varieties like Fuzhou and Xiamen, as well as 'Hakka' (one assumes Meixian) have only [ɛ] or [e]. In Norman's (1988) overview of Mandarin, one can see that while geographically central varieties like Yangzhou have [ɤ], southern ones like Chengdu have [e]. Xi'an and Wuhan both have [ɤ] (Sun 2007; Zhang 2015). Shanghai Wu (Zee and Xu 2017) has both [ɛ] and [ə], while Suzhou has [e] (Handel 2017). Nanchang Gan (Coblin 2015) has both [e] and [ə]. Finally, as will be shown for Xiang, Changsha has only [ɤ], Shaoyang has [ɛ], and Hengyang has both. We therefore see the familiar three-way division portrayed once more: northern and southern extremes with a central transitional zone (see Figure 2).[7] Unsurprisingly, Xiang can be seen to reflect all three.

All of the Xiang dialect areas surveyed have several features in common, one of which is a vocalic inventory of six to seven distinct vowel qualities in open CV syllables (Li 1986 for Hengyang; Wu 2005 for Changsha; Cui 1998 for Yiyang; Zhang et al. 1988 for Taojiang; Chu 1998 and Bao 2017, pp. 130–31 for Shaoyang; Fang 1999 for Yueyang; Lu 2001 for Zhuzhou; Sun 2002 for Shaodong; Song 2006 for Xiangyin; Bei and Shi 2011 and Zhang and Chen 2015 for Shuangfeng; Huang 2012 for Ningxiang; Wang 2013 for Xiangtan; Bao 2017 for Chengbu; Wang 2020 for Yongzhou). The major source of variation in terms of vowel quality is treatment of mid-central/back v. mid-front vowels. As one moves toward the southwest, mid-central and mid-back vowels (primarily schwa) tend to be lost in favor of mid-front vowels (namely [e] or [ɛ]) and vice-versa;[8] this is roughly correlated with the Changyi-Loushao (New-Old) division. There are several exceptions: Xiangtan, Zhuzhou, and Taojiang (New) have only a mid-front vowel, while Ningxiang (New), Shuangfeng

(Old), and Chengbu (Old) have both a contrastive mid-front and mid-central vowel (see Table 1).

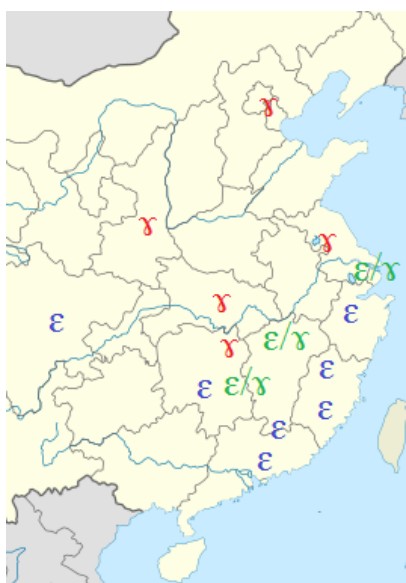

**Figure 2.** Open syllable unrounded mid-vowel contrasts.

**Table 1.** Vowel inventories in open monophthongal syllables in Xiang varieties (north-to-south)[9].

| Dialect Area | High Front | Mid-Front | Low | Mid-Central | High Back | Back |
|---|---|---|---|---|---|---|
| 岳阳Yueyang | [i], [y] | 0 | [a] | [ə] | [u] | [ɤ], [o] |
| 湘阴Xiangyin | [i], [y] | 0 | [a] | [ə] | [u] | [o] |
| 益阳Yiyang | [i], [y] | 0 | [a] | [ə] | [u] | [o] |
| 桃江 Taojiang | [i], [y] | [ɛ] | [a] | 0 | [ɯ], [u] | [o], [ɔ] |
| 宁乡Ningxiang | [i], [y] | [ɛ] | [a], [æ] | [ə] | [u] | [ɔ] |
| 长沙Changsha | [i], [y] | 0 | [a] | [ɤ] | [u] | [o] |
| 株洲Zhuzhou | [i], [y] | [e] | [æ], [ɒ] | 0 | [u] | [o] |
| 湘潭Xiangtan | [i], [y] | [e] | [æ] | 0 | [u] | [o], [ɔ] |
| 衡阳Hengyang | [i], [y] | [e] | [a] | [ə] | [u] | [o] |
| 邵阳Shaoyang | [i], [y] | [ɛ] | [a] | 0 | [u] | [o] |
| 邵东Shaodong | [i], [y] | [ɛ] | [a] | 0 | [ɯ] | [o] |
| 双峰Shuangfeng | [i], [y] | [e] | [a] | [ə ~ ɤ] | [u] | [o] |
| 永州Yongzhou | [i], [y] | [e] | [æ], [a] | 0 | [u] | [o], [ɔ] |
| 城步Chengbu | [i], [y] | [ɛ] | [a] | [ə] | [u] | [o] |

Hengyang, which is a representative member of Chen and Bao's (2007) transitional Hengzhou grouping, resembles Shuangfeng and Chengbu in having both qualities contrasted. We can see that the distribution of vowels is not entirely reflective of the traditional Changyi-Loushao (or New-Old) split, although it is correlated somewhat. The Chen and Bao typology is perhaps a better fit, but it is also not entirely sufficient. We can therefore define three groups of dialects based on presence of contrastive mid-front and mid-central vowels: dialects which have only a central vowel and no mid-front vowel (Changsha, Yiyang, Xiangyin, and Yueyang); dialects which have only a mid-front vowel and no central vowel (Yongzhou, Shaodong, Shaoyang, Xiangtan, Zhuzhou, and Taojiang); and dialects which have both (Ningxiang, Hengyang, Shuangfeng, and Chengbu). These groups will be henceforth referred to as Northeastern (Group 1), Southwestern (Group 2),

and Transitional (Group 3), respectively. These facts are summarized in Figure 3;[10] Group 1 in red, Group 2 in blue, and Group 3 in green, [e] and [ə] also standing in for [ɛ] and [ɤ], respectively.

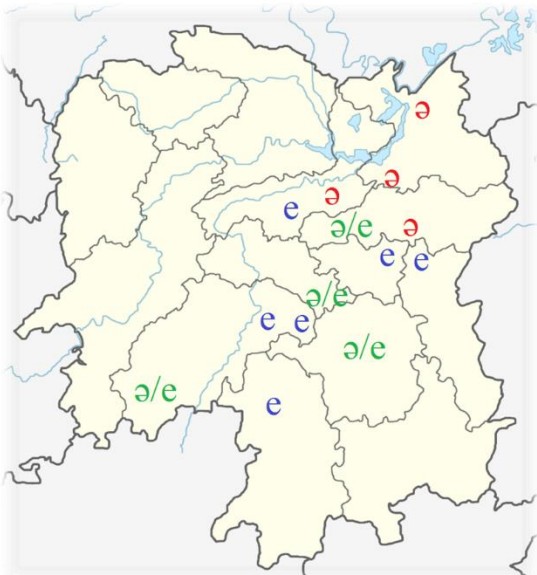

**Figure 3.** Contrastive open syllable mid-vowels by dialect.

　　The choice to focus on mid vowels as a source of typological variation has two motivations: on the one hand, these are reflective of larger northern (Group 1), southern (Group 2), and central (Group 3) phonological trends in the lexical vocabulary of Sinitic and are predicted in some way to reflect larger influences on Xiang. On the other hand, these vowels are particularly important for hesitation markers, which tend toward these vowel qualities across languages (Candea et al. 2005; Dingemanse and Woensdregt 2020); these patterns will be covered in the following section.

### 2.3. Hesitation Markers: Usage and Insights

　　Hesitation markers are an integral, yet often misunderstood, facet of human language. They have clear, essential functions in maintaining naturalistic flow in conversation (Clark and Fox Tree 2002; Dingemanse 2017), but the most common terms for them, disfluencies (e.g., Shriberg 2001; McDougall and Duckworth 2017) and filled pauses (e.g., Gick et al. 2004; Zhao and Jurafsky 2005), seem to indicate that they are sub-linguistic in some sense. The argument made here is that these items are more than simply speech errors or sounds filling empty space and can provide insights into the organization of a language's linguistic system.

　　For the purposes of this study, hesitation markers are defined as minimal discourse markers which serve the primary function of marking a pause in speech (Clark and Fox Tree 2002). This research makes a distinction between 'lexical' and 'non-lexical' hesitation markers; the former includes items which have dual functions as both full lexical items and as discourse markers indicating pauses (English 'like' and 'so'; Cantonese 咁 *gam2* and 就 *zau6*; Ecuadorian Spanish 'y' and 'bueno'; etc.),[11] while the latter includes items which do not have other lexical functions other than indicating pauses (English 'uh' and 'um'; Cantonese 欸 *e6* or 嗯 *m6*; Spanish 'eh'; etc.). One will notice the lack of overt similarities in the former group, contrasted with the form similarities present in the latter group: they are primarily composed of monophthongal vowels and bilabial nasals. The present study is concerned only with the latter non-lexical group, given its strong form correspondences, since the lexical variants are predicted to have largely arbitrary forms, given their dual functions in the lexicon. It is of course very likely that lexical HMs have other types of

form convergence at the suprasegmental level, especially relating to duration and F0, but these will not be addressed in the present piece.

While it is currently understood that non-lexical hesitation markers can tell us a good deal about language, this has not always been the case. Historically these items were often treated as purely non-linguistic (Müller 1861; Chomsky 1965), akin to non-linguistic sounds like grunts. Research by Clark and Fox Tree (2002), however, has shown that far from being non-linguistic, HMs are in fact words of the languages in which they occur, with an identifiable form and clearly defined functions in moderating discourse; Clark and Fox Tree even find that English HMs have different conventionalized forms based on whether they mark a longer or a shorter pause. HMs are in many ways comparable to discourse markers (Maschler and Schiffrin 2015) in that their main function is to facilitate communication. In addition, their status as words means that they have a form that is specific to the language in which they occur; for instance, there is evidence that speakers can identify a language by its HMs (Vasilescu et al. 2005). Fromkin et al. (2014) write, 'Even conversational fillers [HMs] . . . are constrained by the language in which they occur.' This quote indicates two things: on the one hand, that HMs are language-specific rather than identical across languages; and, through use of the word 'even', that they are perceived as having a special status relative to other types of vocabulary.

Despite their language-specific form, hesitation markers are not as arbitrary as most lexical vocabulary. Research on the phonetic form of HMs has shown that these items display a certain degree of convergence across languages (Candea et al. 2005; Dingemanse and Woensdregt 2020), born from environmental pressures in facilitating communication (Dingemanse 2017). These pressures are believed to optimize HMs along several dimensions, including a preference for minimal and non-salient form (monosyllabicity, low pitch), long duration (easily extendable sounds like vowels and nasals), and effort-preservation (bilabial nasals and mid-vowels, as those sounds are most reflective of the vocal tract at rest). These features can be seen to stem from HM functions: they are used to fill pauses in speech without adding to the semantic content of the utterance and need to be easily distinguishable from regular vocabulary without disrupting the flow of communication (Dingemanse 2017; Dingemanse and Woensdregt 2020). A pause is most easily filled with long segments requiring little effort, which would ideally be marked as non-meaningful elements by way of their reduced form, reducing disruption to communication.

Given all of these constraints on optimal HM form, all that is really left to variation is the precise quality of the vowel in question. In a study looking at hesitation marker acoustics in seven languages, Candea et al. (2005) show that languages have a strong preference for mid vowels in HMs, but in each language there is a clustering towards a contrastive quality in said language's phonological inventory. For instance, in English the vowels cluster around [ʌ] (when stressed), but in Spanish they cluster around [e]; while both fall under the mid-vowel category, the precise quality in terms of fronting is a source of variation. The preponderance of mid vowels is believed to be related to pressures of least effort, but the precise quality is expected to be language-specific.

One may wonder at this point why we should focus on HMs and not regular vocabulary which unambiguously reflects the contrastive vowel phonemes of a language. It would be simpler to test the above typology using a lexical item like 车 'vehicle' for instance (which has a mid-vowel in most of the dialects covered here), since one does not have to first prove that 车 is a word or representative of Sinitic phonology. The short answer is that while the mid-vowel split in Sinitic can be proven using lexical vocabulary, whether this can be done with hesitation markers (or other potential sub-lexical material) has not been established. The purpose of the present piece is not to prove mid-vowel variation in Sinitic, which is readily observable, but to explore whether this type of variation can or cannot be expanded to sub-lexical vocabulary like HMs, which are relatively unexplored in Sinitic. In addition, the ultimate goal is to determine whether HMs can or cannot be used as evidence for typological splits, despite their special status. It will be seen in

Section 4.2 that while equivalent lexical vowels and HM vowels do differ in systematic ways, they display comparable acoustic correlates for contrasts.

An additional concern still unaddressed is that many of the studies above address HM trends for a limited set of languages (e.g., Candea et al. 2005; Vasilescu et al. 2005; Dingemanse and Woensdregt 2020), meaning there is insufficient evidence to assume that varieties of Xiang will display similar patterns. For instance, while we may be able to determine a consistent vocalic quality for Mandarin hesitation markers, as Candea et al. (2005) do, this does not necessarily prove that we should be able to do this for Xiang (or any other language not addressed by the authors), despite its close genetic relationship with Mandarin. However, by the same token, there is no reason to assume Xiang is unique in this respect, as the existing evidence seems to point towards there being identifiable qualities in HM vowels, which can be distinguished from language to language, and there does not appear to be evidence of a language employing exclusively non-inventory vowels in its HMs (as of yet).

Given the variation in terms of Xiang contrastive mid vowels covered in Section 2.2, and the tendency towards centrality that HM vowels display, one wonders whether HMs will reflect the typological split of mid vowels in Xiang. This then leads to the question: based on what we know about cross-Sinitic and cross-Xiang typology and phonological features, can the phonetic/phonological form of hesitation markers serve as additional evidence for traditional boundaries? Or will they perhaps indicate a different kind of divide, or no divide at all? An attempt at addressing these questions will be conducted through the study covered in the following section.

## 3. Methodology

The data for this research were gathered through an online experimental platform called Gorilla (www.gorilla.sc, first accessed 16 May 2021), which allows experimental procedures to be conducted through an online link (Anwyl-Irvine et al. 2019). Speakers were recruited through WeChat (Tencent Holdings Limited 2022) and word of mouth. Speakers were required to perform the Pear Stories Paradigm (Chafe 1980; Erbaugh 2001, 2013), wherein a speaker relates the events of a video about a child stealing pears. The speaker was instructed to record themselves on a built-in laptop or smartphone microphone at the following website: https://addpipe.com/simple-recorderjs-demo/ (first accessed 5 January 2021), and then return the recording to the researcher. All phonetic analysis was conducted in Praat (Boersma and Weenink 2022) by hand. This included taking F1 and F2 from the steady-state portion of the vowel as an average; F0 and duration were extracted as well. Datasets associated with this project can be found at either of the following links: https://doi.org/10.25442/hku.20652330 (created 21 August 2022) or Supplementary Materials.

The experimental paradigm defined above is predicted to effectively elicit HMs, since it involves cognitive effort and information retrieval. The highest number of self-initiated hesitation markers are predicted to surface as informants attempt to recall the events of the video they have just watched; Erbaugh (2001, 2013; http://www.pearstories.org/, accessed on 21 January 2021), for instance, manages to elicit these items in abundance in her own application of this paradigm to Sinitic.

One potential issue with this type of methodology is the proliferation of different recording devices used by speakers to record themselves, which can cause issues of comparability across devices (Sanker et al. 2021). The recordings took place on three types of devices: laptop microphones (n = 24), smartphones (n = 21), an iPad (n =1), and the HKU sound booth (n =1). Of particular concern here is the potential effect of file compression on the resulting recordings, relevant for smartphones and iPads. For laptops, this is avoided by instructing the informant to record themselves on non-compressing software (https://addpipe.com/simple-recorderjs-demo/, first accessed on 5 January 2021), which outputs WAV files. However, Freeman and DeDecker (2021) and Zhang et al. (2021) recommend usage of smartphones for remote research, as these devices (as well as iPads)

tend to use lossless compression of audio files in order to avoid unnecessary distortion and are therefore often reliable. It should be noted that there is still some lossless compression relative to WAV files, but that they are still much preferred to MP3 files, for instance. An additional concern might be the difference in quality between microphones (especially laptop-internal v. external microphones; see Calder et al. 2022), which cannot be verified with this particular methodology. However, it should also be mentioned that no systematic differences could be observed on vowels recorded on smartphones vs. laptops. While Sanker et al. (2021) recommend against comparing across different device types, they also point out that smartphones and iPads do not seem to alter vowel formants in significant ways, and they point out that acoustic correlates of phonological contrasts are still recoverable from the signal, despite certain inconsistencies.

### 3.1. Xiang Varieties in the Sample

The present sample gathers data from 47 speakers representing 16 Xiang varieties; this is summarized in Figure 4 (outline of Hunan: "Location Map of Hunan, China", Wikimedia Foundation. 2014. Location Map of Hunan, China, https://creativecommons.org/licenses/by/4.0/deed.en, accessed on 21 November 2021); locations and names added by author).

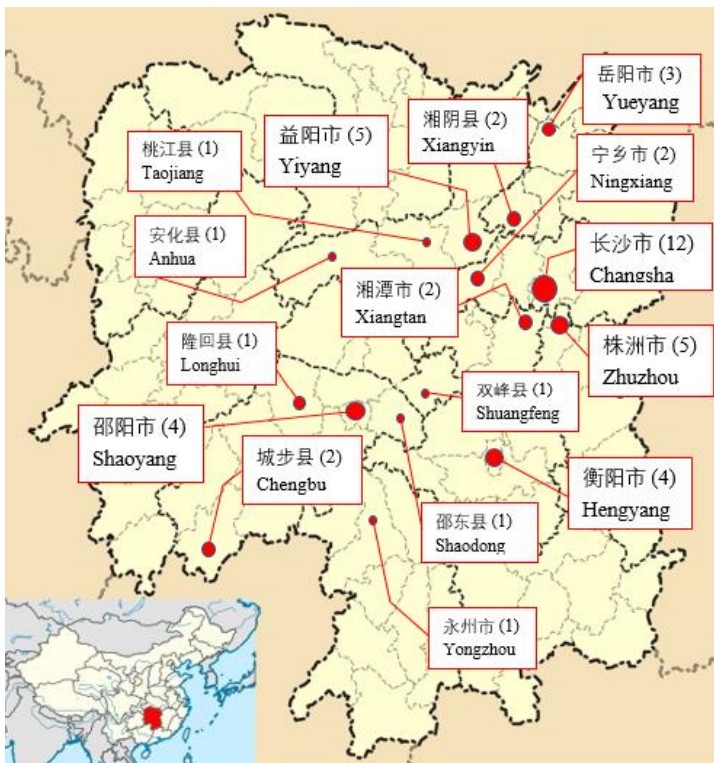

**Figure 4.** Dialect areas represented by the sample (no. of speakers in parentheses).

In terms of the traditional classification, eight of the dialects are New Xiang (36 speakers) and seven are Old Xiang (11 speakers). Following Chen and Bao's (2007) classification, the dialects fall into four groups: seven are Changyi (32 speakers), six are Loushao (10 speakers), one is Yongquan (1 speaker), and one is Hengzhou (4 speakers). If any of the dialects in the sample lacked hesitation markers, they were excluded from the analysis; see Table 2.

**Table 2.** Speakers and Dialect Areas in the Sample.

| Dialect Area | No. of Speakers | Typo. Affiliation | Chen and Bao (2007) | No. of HMs |
|---|---|---|---|---|
| 岳阳市Yueyang | 3 | Changyi-New | CY | 41 |
| 湘阴县Xiangyin | 2 | Changyi-New | CY | 21 |
| 益阳市Yiyang | 5 | Changyi-New | CY | 35 |
| 桃江县Taojiang | 1 | Changyi-New | CY | 4 |
| 宁乡市Ningxiang | 2 | Changyi-New | CY | 9 |
| 长沙市Changsha | 12 | Changyi-New | CY | 92 |
| 株洲市Zhuzhou | 5 | Changyi-New | CY | 4 |
| 湘潭市Xiangtan | 2 | Changyi-New | CY | 4 |
| 衡阳市Hengyang | 4 | Changyi-New | HZ | 21 |
| 安化县Anhua | 1 | Loushao-Old | LS | 0 |
| 邵阳市Shaoyang | 4 | Loushao-Old | LS | 31 |
| 隆回县Longhui | 1 | Loushao-Old | LS | 0 |
| 邵东县Shaodong | 1 | Loushao-Old | LS | 3 |
| 双峰县Shuangfeng | 1 | Loushao-Old | LS | 20 |
| 永州市Yongzhou | 1 | Loushao-Old | YQ | 3 |
| 城步县Chengbu | 2 | Loushao-Old | LS | 5 |

Two dialect locations, Anhua and Longhui, were excluded for lack of HM tokens (each with only a speaker apiece), bringing the total number of dialects analyzed down to 14. Given the low number of speakers per dialect location, analysis is done by subgroup (see Sections 2.1 and 2.2), rather than by individual dialect. Demographic information for all speakers is covered in the following section.

*3.2. Demographic Information*

The female-to-male ratio is 33/14, and the average length of recordings is 128 seconds (37–264 s). The average age of the informants is 26.5, with a range between 18–53. In terms of education, 27 of the respondents had a master's degree or above, 19 had a bachelor's degree, and only 1 had a high school diploma. The informants are generally younger and more extensively educated than what would be considered ideal for dialect research; younger and highly educated speakers are often more mobile and may be more influenced by Putonghua (Kurpaska 2010, p. 134). All speakers are in fact bilingual in PTH; however, all participants purport to be speakers of the dialects in question, with varying degrees of self-reported fluency coded on seven levels: Native, Near-Native, Fluent, Conversational, Average, Basic, and None. No respondents replied below 'Conversational'; speakers who self-reported below 'Near-Native' were placed in a 'heritage speaker' category, which is assumed to be heavily PTH influenced (n = 9). The PTH vowel paradigm in this case is identical to the Northeastern or Changsha paradigm, with only a mid-back or central quality distinguished in open syllables. The ubiquitous presence of PTH bilingualism is predicted here to lead not only to the adoption of PTH features in the heritage group, but also to result in possible interference in higher fluency speakers, and the potential overgeneralization of the NE or Changsha paradigm. This is discussed as a potential confounding factor in Section 5.

There was a wide divergence in terms of how many hesitations each speaker produced, which is in line with previous research on the high speaker-specificity of rate of HM usage (Braun and Rosin 2015; McDougall and Duckworth 2017). The average of 6.5 HMs per respondent hides the fact that there was a range of 0 to 39 hesitations; 7 speakers had no hesitations whatsoever, and one speaker had only syllabic nasal hesitations; both groups were removed from the analysis.

It should be mentioned as well that many speakers are multilingual not just in PTH, but also in other varieties of Xiang. At least six speakers report some degree of fluency in another variety of Xiang, usually from regions that border their hometown. For instance, XT1 (Xiangtan) is also fluent in the Zhuzhou and Changsha dialects, which are only sepa-

rated by a few kilometers. Others are proficient in more distant varieties; CB2 (Chengbu) is proficient in the Shaoyang and Changsha dialects, which are far removed. This is only predicted to result in complications when the varieties a speaker is proficient in have a different mid-vowel paradigm than their native dialect. For CB2 and most other multi-dialectal speakers this is unproblematic, as most contiguous varieties have similar paradigms and Chengbu distinguishes both qualities analyzed here. However, for XT1 for instance, who reports fluency in Changsha, this may lead to interference; this will be discussed further in Section 5.

## 4. Results

This section covers descriptive statistics including HM syllable structure and average F0 (Section 4.1) and then moves on to describe HM vowel quality (Section 4.2) and compare its distribution against the two proposed dialect groupings (Section 4.3). Finally, a look at the distribution of HMs within each grouping is presented in Section 4.4.

### 4.1. Overview

The recordings resulted in a collection of 304 hesitation markers (167 from females), which came in a variety of syllabic formats, although the majority (n = 215) were simple monophthongs. The second most common were VN (n = 35) and N (n = 23). In total, there were 10 different recorded syllabic formats (see Figure 5). The only syllable type excluded from the analysis was N, since the focus is on vocalic formant structure. HMs were largely isolated from surrounding speech, with a pause between the HM and an utterance either preceding (158), following (7), or both (130). Only a very small number (9) occurred in connected speech.

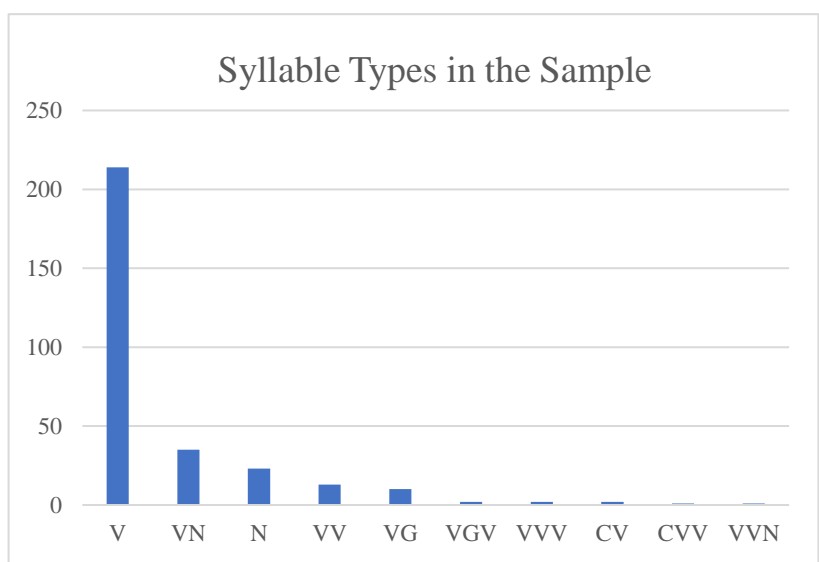

**Figure 5.** HM syllable types.

In addition to the commonly seen N and VN formats (most often approaching [ŋː] and [əːm]), HMs can sometimes be followed by an offglide (both palatal and labio-velar, approaching [əːj] or [əːw]) or be diphthongs with changing vocalic quality by the end of the syllable (approaching [aːə] or [eːə], for instance). Since HMs can be very long (average duration = 435.93 ms), it is reasonable that a single vocalic quality is not maintained throughout the entire syllable; there were even two instances of triphthongs (approaching [aːəːa]). There were at least three instances of initial consonants in the sample, all of which were laryngeal fricatives (approaching [haː]). The only constant, it seems, is that all HMs were composed of a single syllable. However, it should be clear from Figure 5 why the choice has been to focus on monophthongal vowels: they are by far the most common

type of HM, accounting for two-thirds of the sample. If more than one vowel quality was identified in a token, only the first vowel in the sequence was retained for analysis.

In terms of F0, the sample was generally in line with crosslinguistic trends for low fundamental frequencies in HMs (Shriberg 2001; Braun and Rosin 2015; Yuan et al. 2016). The average female F0 was 169.3 Hz (standard deviation = 32.7), with a range between 67.1–223.1 Hz, while the average male F0 was 111.2 Hz (standard deviation = 18.9), with a range between 53.2–148.5 Hz. HM F0 contours were extracted at 12 equidistant points using a Praat script written by Jon Havenhill, and are plotted below in Figure 6.[12]

The female F0 data represent a steady contour averaging around 170Hz throughout, sloping slightly downward near the end of the syllable; the average is below the predicted overall average F0 of 220 Hz for females (Reetz and Jongman 2020). The male contour represents more movement, due to the high level of variation in this group,[13] and averages about 125–130 Hz, which is about the overall F0 average for males (Reetz and Jongman 2020).

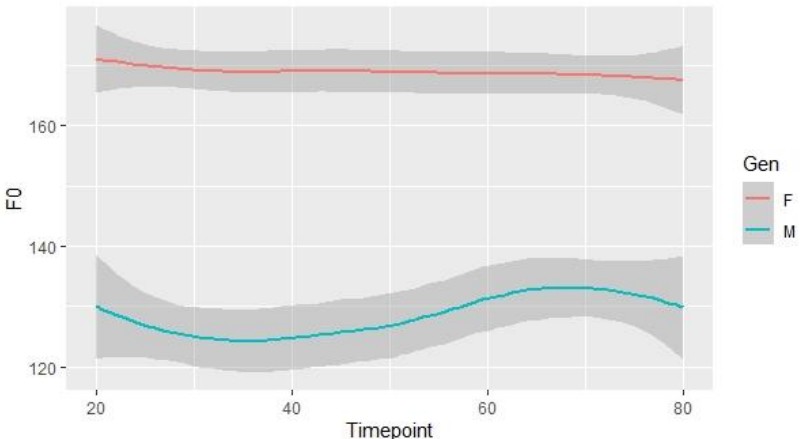

**Figure 6.** HM F0 contours (female = red, male = blue).

## 4.2. Vowel Quality

Four distinct HM vowel qualities were identified in the sample, coded as: front (40), central (115), back (69), or low (57), based on the author's transcription. The choice to group HMs into these particular categories is based on observations summarized in Table 1: most varieties have six contrastive qualities (ignoring rounding), reduced to four when high vowels are removed (note below the general absence of high vowels in the sample). This assumes that an HM vowel is at least analogous to an acoustic equivalent in the phonemic vowel inventory (following work in Candea et al. 2005); for instance, that a front-quality HM vowel is comparable (if not perhaps identical) to a Xiang variety's /e/ ([ɛ] or [e]). This assumption is not entirely without issue; as will be seen below, there are some acoustic divergences between lexical and HM vowels.

F1 and F2 were taken as means across the steady-state portion of each vowel, with each selection between 99 and 50 ms. It was observed that HMs tended to fall within the mid-to-low acoustic space (overall mean = F1 657.2 Hz, F2 1470 Hz), rarely rising above the mid height or being excessively fronted (with only two HMs below an F1 of 350 Hz, and two above an F2 of 2250 Hz); this is summarized in Figure 7.

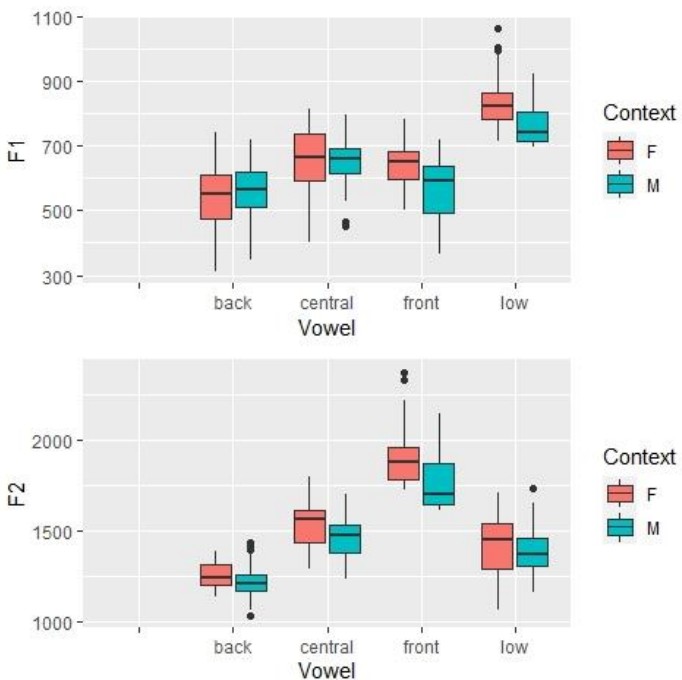

**Figure 7.** Average F1 and F2 of sampled vowel qualities (female = red; male = blue).

Average F1 and F2 by the different vowel qualities identified can be summarized in Figure 7. There were no instances of HM vowel quality outside of the mid-to-low range; that is to say, there were no high vowels in the sample. These assumptions can also be corroborated quantifiably, as seen in Figure 8.

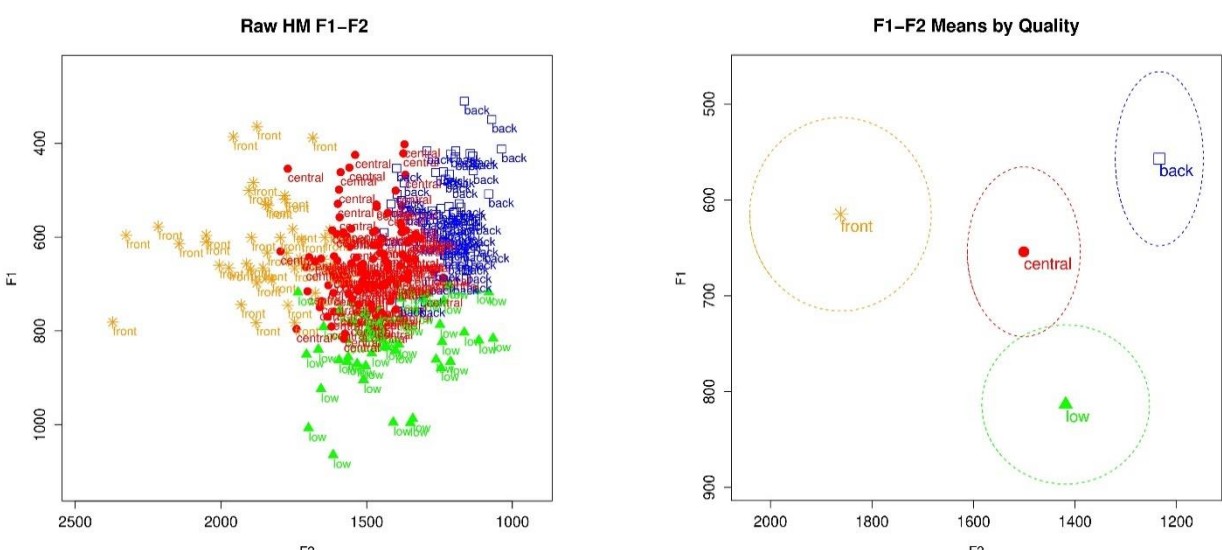

**Figure 8.** Vowel tokens (**left**) and means (**right**) in the sample (all speakers).

The mid-back quality is most likely a variant of the central quality, and was distinguished primarily based on the author's transcription, considering that only Yueyang has a phonemic contrast between the two (Fang 1999). Since Yueyang contributes the second most HMs to the sample (n = 41), it is perhaps unsurprising that this quality was discernible. For varieties that do not distinguish these, it is possible that one quality or the other is prioritized in each dialect in open syllables, as, for instance, is the case for Changsha (Wu 2005, pp. 371–72) or Putonghua (Duanmu 2007, p. 37), where the mid-back quality is restricted to open syllables and the central quality to nasal-final syllables(e.g., [+vocalic, +mid] →

[+central]_N\$; [+vocalic, +mid] → [+back]_\$). In reading these data, therefore, these two should be treated as largely interchangeable, or at least as constituting a diffuse boundary in the vast majority of dialects.

If all speakers in the sample are pooled together, one can get an idea of the distribution of HMs in terms of F1 and F2 (see Figure 8).[14] As pointed out above, HMs were grouped into four main qualities, under the assumption that they are analogous to phonemic qualities in the varieties in questions.

As Figure 8 shows, there is a division of the vowel space between the four qualities identified, although there is a degree of overlap between the 'central' and 'low' qualities. It is important to note that only Yueyang (Fang 1999) makes a phonemic distinction between mid-central [ə] and mid-back [ɤ], and that dialects like Changsha (which contributes the most HMs to the sample) have these qualities as contextually conditioned allophones. It is also worth pointing out that low fluency heritage speakers in the sample are assumed to pattern more closely with PTH, which has [ɤ] as the only permissible quality in open syllables (Duanmu 2007, p. 37).[15] The mid-front and mid-central/back qualities, as the focus of the present study, will be discussed further below.

Since the front and central HM qualities are the ones crucial to the analysis, they were also compared to lexical vowels extracted from the root 车 'vehicle' (pronounced either [tsʰe] or [tsʰɤ]/[tsʰə] depending on the contrast active in a variety), which occurs often (37 speakers have at least one instance) in the narratives in reference to a bicycle occurring onscreen in the video. This was done in order to test whether this contrast in the HMs could also be recovered from the lexical vocabulary. The results can be observed in Figure 9; the central and back qualities are combined in the 'hmc' category, as these are likewise not distinguished in the lexical vocabulary, and the low quality is excluded as it is not a direct focus of the present study.

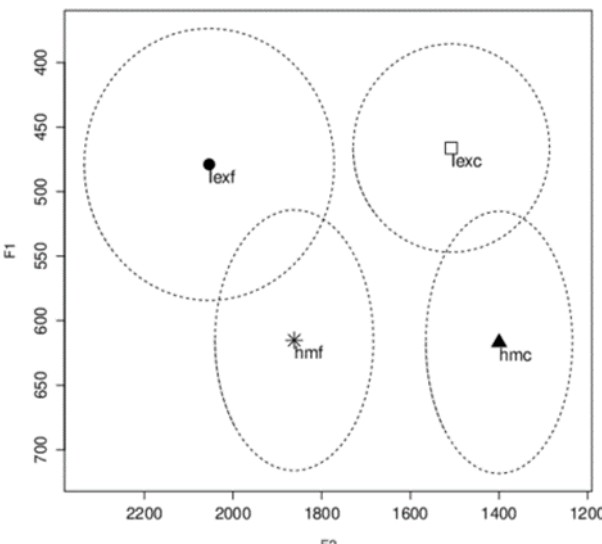

**Figure 9.** Means for front and central quality vowels, for both lexical and hesitation marker vowels ('lexf' = front lexical vowels; 'lexc' = central/back lexical vowels; 'hmf' = HM front vowels; 'hmc' = HM central/back vowels).

The means in Figure 9 do not overlap, with lexical vowels from 车 being considerably higher and more fronted on average, unlike in previous findings on HM vowels (Sevilla 2021), which found rather direct overlap between vowel categories. The difference may be due to differing contexts: the lexical vowels used here occur after a coronal affricate in every instance, while HMs occur in isolation. T-tests on the means indicate substantial differences between the two sets, with all p-values highly significant (see Appendix A

Tables A1 and A2); HM vowels appear to be consistently less fronted and lower than their lexical counterparts. This raises questions about the tenability of comparison across lexical and hesitation marker vowels: are they really instances of the same phonological vowel? However, Dingemanse (2017) and Dingemanse and Woensdregt (2020) mention that HMs are particularly susceptible to ease of articulation considerations in ways in which lexical vocabulary is not; that HMs should be considerably more centralized than their lexical equivalents is therefore perhaps not surprising. What is key to the present study is that in each case, despite the lack of direct overlap, the acoustic correlate of the contrast is maintained in each pair: a front quality can be distinguished from a central quality through an increase in F2, and both qualities are comparable in terms of F1 within each pair. A front-quality HM is therefore predicted to be perceived by speakers as more [e]-like than a central-quality one, if not identical.

### 4.3. Vowel Quality by Typological Affiliation

Turning now to typological affiliation, three groups are analyzed following Chen and Bao's (2007) classification: Changyi, Loushao, and Hengzhou. The data for Yongzhou (which would fall under the fourth group of Yongquan) are incorporated into Hengzhou for lack of tokens. A fourth 'Heritage' grouping is also included, composed of speakers with low self-reported proficiency (see Section 3.2). This does not represent any sort of typological region but is rather included for comparison. The dialects represented by the nine speakers in this group are as follows: Changsha (n = 4), Xiangyin (n = 2), Hengyang (n = 2), Zhuzhou (n = 1); Chen and Bao: Changyi (n = 7), Hengzhou (n = 2). The results can be observed in Figure 10.

**Figure 10.** Mean F1-F2 by affiliation in Chen and Bao (2007), incl. heritage speakers.

In Figure 10 the different shapes, as well as the colored circles, represent different vowel qualities identified in the sample. The star represents the 'front' vowel quality (approaching [e] or [ɛ]); the filled dot represents the central quality (approaching [ə]); the empty square represents the back quality (approaching [ɤ]); the filled triangle represents the low quality (approaching [a]). Figure 10, based on the work of Chen and Bao (2007), shows that HM vowels do not display any particular taxonomic clustering; this can be seen from the fact that all groupings have both mid-front and mid-central qualities, and in fact have all four qualities represented. However, the fact that all of the Chen and Bao groupings have a mix of qualities is in line with the status of Xiang as transitional between larger northern and southern Sinitic groups, which more clearly pattern one way or the other.

It is noteworthy that compared to the other three qualities, which cluster very closely together, the front quality demonstrates a larger variation in values. For instance, the front quality for Heritage is further back than the other three groups (avg. F2 1710 Hz), while Loushao is dramatically fronted (avg. F2 of 1954Hz), although the small number of items for the former (n = 5) makes it difficult to conclude anything on this point. At the same time, there is also the unexplained exaggerated height of the Hengzhou and Changyi groups (avg. F1 of 512 and 570 Hz), further highlighting the differences between the four groupings. However, this range of values still falls roughly within the expected range for a mid-front vowel (using Cantonese [ɛ] and [œ] as references; Shi et al. 2015), so it is unclear whether this variation is not simply due the averages being influenced by idiolectal variation or a gender imbalance between the two groups. For instance, Loushao has a higher proportion of female speakers (7/9) compared to Changyi (15/22), which is predicted to result in higher frequencies (Hillenbrand et al. 1995).

The failure of HM vowel quality to support the original typological scheme is unsurprising given the variability within each group in terms of mid vowels in CV syllables. If we instead consider the alternative typology based on mid vowel quality (proposed in Section 2.2), we can re-organize the dialects into three groups:

1. Northeastern (Group 1): mid-central vowel only (Changsha, Yiyang, Xiangyin, Yueyang)
2. Southwestern (Group 2): mid-front vowel only (Yongzhou, Shaodong, Shaoyang, Xiangtan, Taojiang, Zhuzhou)
3. Transitional (Group 3): both mid-central and mid-front vowels (Ningxiang, Hengyang, Shuangfeng, Chengbu)

To these are added the previous heritage (HT) group as well; the nine speakers here are split into NE (n = 6), TR (n = 2), and SW (n = 1). As all speakers are native speakers of Mandarin only, the HT group is expected to pattern most closely with the NE pattern (which Mandarin shares). However, non-Mandarin patterns were observed to emerge for one of the HY speakers (see below).

The results are summarized in Table 3. The key point of note is that once again, all three of the proposed groupings share all four of the observed HM vocalic qualities; while some may have more or fewer (for instance, NE and HT have many fewer instances of 'front' HMs and SW has many fewer instances of 'back' HMs), it is not possible to discern any clear division between the groups (but see Section 5.1 for discussion).

**Table 3.** Summary of results by alternative grouping.[16]

| Area | Quality | No. of Tokens | Average F1 | Average F2 |
|---|---|---|---|---|
| NE | central | 44 | 667.5 | 1473.8 |
| NE | back | 27 | 570.9 | 1213.9 |
| NE | low | 23 | 770.0 | 1370.7 |
| NE | front | 12 | 551.1 | 1814.8 |
| SW | central | 13 | 661.7 | 1551.3 |
| SW | back | 2 | 628.6 | 1233.5 |
| SW | low | 16 | 816.8 | 1451.9 |
| SW | front | 12 | 652.1 | 1933.5 |
| TR | central | 25 | 644.2 | 1502.8 |
| TR | back | 8 | 567.4 | 1281.4 |
| TR | low | 5 | 914.8 | 1319.6 |
| TR | front | 10 | 625.9 | 1923.8 |
| HT | central | 33 | 641.4 | 1515.7 |
| HT | back | 32 | 538.6 | 1239.0 |
| HT | low | 13 | 848.0 | 1499.6 |
| HT | front | 6 | 651.5 | 1710.0 |

These results can be more easily observed in Figure 11, which shows the position of the averages relative to each other. One should keep in mind that the imbalanced nature of

the number of tokens (given the small sample size) is a confounding factor here. However, once again as in Figure 10, we have a clustering of vocalic qualities into four groups.

**Figure 11.** HM vowel means grouped by mid-vowel contrasts, incl. heritage speakers (star = front; filled dot = central; empty square = back; filled triangle = low).

Particular differences between Figures 10 and 11, the most conspicuous of which is the low quality (represented with a black triangle) being somewhat higher in Figure 11, are largely unexplained. Meanwhile, the wide divergence between the groups in terms of the front quality is still apparent (with NE still higher than other groupings), although the TR and SW groups now share a more fronted quality than the other two. Once again, these fall within the expected range for a mid-front vowel and can be potentially attributed to gender imbalances in the data.

### 4.4. A Closer Look at HM Incidence by Grouping

While the results in Section 4.3 show that the groupings cannot be distinguished in a categorical way using HMs, a closer look at the incidence of each vowel quality within each group suggests that an approach looking at gradient distinctions may be more successful. Consider the data presented in Table 4, which show the proportions of each vowel quality by group.

**Table 4.** Vowel quality proportions by group (open syllable mid-vowel contrasts).

|  | No. of Items | Northeast | Southwest | Transitional | Heritage |
|---|---|---|---|---|---|
| **Front** | 40 | 0.11 | 0.28 | 0.21 | 0.07 |
| **Central** | 115 | 0.42 | 0.30 | 0.52 | 0.39 |
| **Back** | 69 | 0.25 | 0.05 | 0.17 | 0.38 |
| **Low** | 57 | 0.22 | 0.37 | 0.10 | 0.15 |
| **Overall** | 281 | 0.38 | 0.15 | 0.17 | 0.30 |

By far, the most common quality is central, which is perhaps unsurprising given potential least-effort considerations in HMs, which might prioritize central quality as that diverging the least from the articulators at rest. The Northeast or NE group provides the highest proportion of HMs overall at 0.38 of the sample, while the smallest proportion belongs to the Southwest or SW grouping at 0.15.

It is telling that the proportion of front-quality HMs is smaller for the NE group than for the SW and TR groups, despite NE hesitations making up about 0.38 of the entire sam-

ple. This distribution is attributed to the fact that front vowels are in any case much less common in this group generally as they do not occur as a distinctive open syllable quality (the fact that they occur at all is what is surprising). Front-quality HMs make up only 0.11 of the NE sample (the smallest of all four qualities for this grouping), while they make up 0.28 and 0.21 of the SW and TR samples, respectively (consider that these two groups together making up only 0.32 of the total HM sample). It is no surprise that SW, with front quality mid-vowels as the only contrastive (un-rounded) mid-vowel allowed in open syllables, has the highest proportion of front quality HMs of any of the other groupings. At the same time, again unsurprisingly given its high input from Putonghua, the HT grouping also has the lowest number of front quality HMs at 0.07, and the only front items provided here were all from a heritage speaker of Hengyang (a TR variety). For front-quality HMs, the direction of all of these incidences is in line with previous assumptions about the phonologies of the dialects making up these groupings, although they are not absolute as would be expected.

On the other hand, however, central-quality HMs make up a comparable proportion when compared to front-quality HMs in the SW group, which is completely unexpected in a group where this quality is not found in open syllables; this can be potentially attributed to crosslinguistic HM pressure to converge on schwa or to influence from other dialects (particularly Mandarin). Central-quality vowels are the most common quality for NE at 0.42, which is unsurprising given that central vowels are the only contrastive unrounded mid-vowel occurring in non-HM open syllables for this grouping.

The back quality is particularly compelling as it betrays significant Mandarin influence, given that this variety only allows the back quality vowel in open syllables (Duanmu 2007, p. 37). The back quality was much less common in the SW/TR groups (0.05/0.17) compared to the NE/HT groups (0.25/0.38), which is expected if these first two groups can be thought of as evincing the least Mandarin influence. In fact, the HT group has the highest proportion of back-quality HMs of any grouping, which is expected given these speakers' first language is PTH.

As we can see, for the most part, the relative incidence of the different qualities is in line with the assumptions gleaned from the literature in Sections 2.2 and 4.2 and does allow one to draw approximate distinctions. The sole exception seems to be the central-quality being over-represented in the SW grouping, despite its non-contrastive status in dialects making up this grouping; this remains unexplained, but may be related to interference from other varieties of Sinitic (PTH or otherwise). These results, the non-categorical nature of the distinctions between groups, and the finding that all four qualities can be found in all three investigated groupings can be interpreted in various ways; some of the most relevant are discussed in following sections.

## 5. Discussion and Conclusions

Of the four HM vowel qualities identified in Section 4.2, all four can be identified in each of the groups outlined above in Section 4.3, which questions the usability of hesitation marker vowel quality as a criterium for a classificatory framework for Xiang. This applies regardless of whether one explores Chen and Bao's (2007) classification or the alternative classification based on the mid-front v. central contrast.[17] However, if one looks at the proportions of the different qualities by grouping (based on mid-vowel contrasts in open syllables), the picture becomes a bit more complex, indicating that distinguishing between the four groups based on vowel quality should be based on more gradient than categorical criteria.

This section discusses different directions from which to analyze the results obtained in Section 4; Section 5.1 through Section 5.4 discuss potential confounding factors which may have contributed to the results, including (in order): the high idiolectal variation HMs display (Section 5.1), hesitation markers and their relationship to default articulatory setting (Section 5.2), the transitional status of Xiang (Section 5.3), and issues relating to phonology or bilingualism (Section 5.4). The section ends with a brief conclusion (Section 5.5).

### 5.1. High Variability in Hesitation Markers

Certain studies have found that hesitation markers have a high degree of speaker-specificity (Braun and Rosin 2015; McDougall and Duckworth 2017), both in usage patterns as well as phonetic form, which may complicate interpretation of the data. In the sample, this is evidenced primarily by the differing rates of HM usage by speaker (with a range of 0–39 despite performing the same elicitation task), but it is also reflected in the choice of vowel quality by speaker. While the former may be simply a matter of differing lexical selection or cognitive constraints on information retrieval, the latter is somewhat more surprising, and points to another factor which separates HMs from general lexical vocabulary. While pronunciational variations displayed by a single lexical item can usually be attributed to some extraneous factor (contextual changes, regional variation, variation in register, etc.), it is often unclear whether differing vocalic quality in HMs is conditioned by variation displayed by a single lexical item or if it instead represents different lexical items under the HM heading.

In the sample, certain speakers seem to employ a wide variety of vowel qualities; for instance, Speaker CS11 has three vowel qualities represented despite only having six HMs in their recording. Does this mean that speaker CS11 has three different HMs, or are these variations of the same HM? Arguably, if these do represent different lexical items, one should be able to discern different functions for them, as Clark and Fox Tree (2002) do for the English uh and um, but these remain elusive at present. Conversely to the case for CS11, some speakers overwhelmingly prioritize one quality over another. For example, speaker NX2 has only instances of syllabic bilabial nasals, speakers NX1 and YZ1 have only 'low' quality vowels, XY1 and CS12 have only 'back' quality vowels despite having 23 HMs between the two of them, etc. Of course, given the short length of the recordings, it is hard to tell whether these trends for each speaker would hold under increased scrutiny, but they do reflect some of the difficulties in speaker cross-comparability.

Speaker-specific effects may mean that a larger sample of speakers may be required to understand central tendencies, or that speakers should be recorded for longer periods of time to see if trends are consistent. In addition, it is not always clear what the lexical status of HMs is, i.e., whether vocalic variants are variations on the same item or represent different items. However, that these central tendencies exist seems to be corroborated by evidence that speakers can identify a language based on the phonetic properties of its HMs (Vasilescu et al. 2005). In addition, despite the high degree of speaker-specificity, HMs are still restrained by more general cross-linguistic constraints, as seen by the limitation to just four identifiable vowel qualities. Precisely how sharply defined these boundaries are remains to be explored.

### 5.2. Phonemic Vowel Quality or Default Articulatory Setting?

In exploring the quality of HMs, one could perhaps wonder, given their status as items given minimal physical effort in production, whether what we are measuring is phonemic vowel quality or what Honikman (1964) terms 'articulatory setting'. In her terms, this is 'the overall arrangement and manoeuvring of the speech organs necessary for the facile accomplishment of natural utterance', the defaults for which differ from language to language (Gick et al. 2004). One could imagine that HMs represent a default 'rest' position for vowels, which would overlap with the most centralized vowel in a language's inventory. If this were the case, it would invalidate the search for discreet divisions among dialects based on vowel quality, since it is unlikely that each dialect will have a different default setting.

However, findings in Gick et al. (2004) indicate that while articulatory setting may influence the production of HMs (in their terms, 'filled pauses'), they also claim that 'it cannot be the case that filled pauses are simply vocalizations of a language's AS [articulatory setting]'. This finding agrees with the data presented here in that Xiang HMs can have a variety of vocalic realizations within each dialect, rather than a default (such as schwa, for instance). In addition, this agrees with the observation in Clark and Fox Tree (2002)

which sees differing acoustic qualities for English HMs fulfilling different discourse functions. Since HMs have both the status of words as discourse markers, and can have variable form, it seems unlikely that they are simply a function of a default articulatory setting.

*5.3. Transitional Status of Xiang*

The status of Xiang as transitional between northern and southern Sinitic has been well-established (Norman 1988; Wu 2005; Chappell 2015; Szeto and Yurayong 2021), so perhaps it is no surprise that a discreet division between the dialects cannot be identified; certain features might well be diffused across dialect boundaries, especially in items as vulnerable to speaker-specific effects as hesitation markers. For instance, many grammatical features such as Xiang classifier systems (Sevilla 2023), disposal and passive constructions (Wu 2005), or aspect marking Wu (1999) display influences from both the north and south and do not cluster into clearly delineated regions. The traditional view of Xiang as neatly divided into northern and southern regions is based on a handful of linguistic features or a single phonological feature, as in the case of Yuan (1960). It is therefore possible that the attempt at exploring these neat divisions was flawed from the beginning.

In a larger sense, what is most remarkable about the quality of Xiang HMs is how they reflect the 'transitionality' of the grouping, with its mixture of northern and southern features and the resulting linguistic diversity of the grouping. This diversity is evidenced by the observation that Putonghua[18] and Cantonese tend to have at most one or two primary HM qualities (Candea et al. 2005; Yuan et al. 2016; Sevilla 2021), whereas Xiang has four. Hunan province, as an area of contact and convergence between typologically distinct varieties of Sinitic, can be seen to exhibit a high degree of diversity, even in its sublexical vocabulary.

As mentioned in Section 2.2, it is important to note that the mid vowels explored in this study do appear to reflect larger trends in Sinitic, even if they do not reflect smaller divisions in Xiang. This could be shown not only by consulting grammars of the varieties in question, but also by exploring the phonetic quality of lexical items, such as 车 'car, vehicle' for instance. The mid vowel split between Cantonese and Mandarin presented in Shi et al. (2015) may prove a further area for exploration of typological variation outside of Xiang. In any case, while HMs may be subject to too many extraneous factors (e.g., least-effort pressures, language-specific phonotactic/phonological restrictions, speaker-specific effects) to serve efficiently as bearers of typological variation, this does not necessarily mean that the distribution of mid vowels in the lexical vocabulary cannot inform Sinitic typology, although this would require a more extensive study.

*5.4. Residual Issues*

One of the core assumptions of the present piece is that hesitation marker vowels are comparable to acoustically equivalent phonemic vowels from the lexical vocabulary. As mentioned in Section 4.2, there are reasons to question this view, as HM vowels were consistently less fronted and lower than their equivalents. As mentioned, this can potentially be attributed to ease of articulation considerations (Dingemanse 2017; Dingemanse and Woensdregt 2020). In the view of the author, the most clear-cut way to determine whether HM vowels do in fact latch on to phonemic equivalents is to conduct a perceptual study where informants are requested to match HM vowels to their own phonemic categories; this is beyond the scope of the present piece but presents fertile ground for further research.

While it is the case that these dialects display the aforementioned restrictions on mid-vowels in open syllables, the image is considerably murkier if one ignores phonotactic restrictions. For instance, while Changsha has [ɤ] in open syllables, it also allows [ɛ] as long as it is preceded by a palatal glide (Wu 2005, pp. 371–72). Therefore, while this vocalic quality has contextual restrictions on its appearance which might theoretically exclude it from occurring in HMs (assuming HMs are subject to similar phonological restrictions as other vocabulary), this vocalic quality is still found in the Changsha dialect and might therefore occur. The reverse situation holds in Shaoyang (Chu 1998), for instance, with schwa occur-

ring in nasal final syllables and [ɛ] occurring elsewhere. This raises questions as to whether or not HMs display the same sorts of phonotactic restrictions as lexical vocabulary, or if they have special status; for the purposes of this study, they are assumed to have the same sorts of phonotactic restrictions as lexical vocabulary, minus their convergence in form due to communicative constraints (Dingemanse 2017), but this view may be complicated through future studies.

An additional issue of some concern which may have influenced the results is the amount of Putonghua exposure of the speakers. As mentioned, nine of the speakers fell below the benchmark to be considered fully native speakers of the dialects in question and were in fact only native speakers of PTH. Compounding this issue is the fact that all of the native Xiang speakers are either balanced bilinguals in PTH or consider it an additional mother tongue; given that all speakers have high familiarity with Putonghua, it is perhaps no surprise that their hesitations are comparable across dialect boundaries. At the same time, at least six speakers reported fluency in other varieties of Xiang, which may further complicate matters. It may be that when switching from one dialect to another, speakers actually change their hesitations; if, for instance, they have a PTH-like accent when speaking Xiang, their HMs may appear more PTH-like, or they may consciously alter their HMs based on the variety used in discourse (for instance, consider the non-Mandarin, front-quality hesitations used by the HY heritage speaker). A possible extension to the present study could involve an exploration of code-switching as it relates to HMs, although this would require a more precise definition of HM form within each individual dialect, rather than across an entire subgroup as was done in the present study.

*5.5. Final Remarks*

The present study has been an attempt at using the feature of HM vowel quality for defining typological regions in the Xiang grouping of Sinitic. The motivations for this research are born out of two deficiencies: a lack of research on HMs in Sinitic and a lack of experimental research on the Xiang grouping outside of traditional dialectology or reading tasks. To this end, a novel three-way typology was introduced based on contrastive mid vowels in open syllables, which divided dialects into those with only mid-front, those with only mid-central/back, and those with both qualities, which is also partially reflective of larger regional tendencies in Sinitic. This was done to test whether HM vowel quality could be mapped onto the groups defined in this way. While the results do not support existing or novel regional divisions in an absolute, categorical way, the relative incidence of HM qualities does seem to support the novel taxonomy, albeit somewhat harried by a host of potential confounding factors.

In addition to the above, the present research contributes to our understanding of the place HMs occupy in the phonological system; specifically, this research queried whether HM acoustic qualities can be mapped onto phonological categories. It was noted that while the front and central contrast was preserved in both cases, there was not a direct overlap between the categories (unlike what was found in Sevilla 2021 for Cantonese). Further research into the space between phonetics and phonology that HMs occupy may help elucidate this disconnect between the two findings.

One potential area of further interest is the distribution of mid vowels in open syllables across Sinitic generally. Future research could address the distribution of these vowels in items with a less controversial lexical status, as well as pursue exploration of Sinitic hesitation markers with a larger sample of speakers to avoid the confounding effect of high idiolectal variation. It is hoped that this study will provide fertile ground for the continued exploration of hesitation markers across languages, as well as continued research on Xiang at all levels of analysis.

**Supplementary Materials:** The following supporting information can be downloaded at: https://www.mdpi.com/article/10.3390/languages8010073/s1.

**Funding:** This research was funded by Viktorija Kostadinova at the University of Amsterdam (UvA reference no.: 171288) through the Getting Data Working Group (https://gettingdata.humanities.uva.nl/).

**Institutional Review Board Statement:** The study was conducted in accordance with the Declaration of Helsinki, and approved by the Human Research Ethics Committee of The University of Hong Kong (EA210151, 7 April 2021).

**Informed Consent Statement:** Informed consent was obtained from all subjects involved in the study.

**Data Availability Statement:** Datasets associated with this research can be found at the following DOI: https://doi.org/10.25442/hku.20652330.

**Conflicts of Interest:** The author declares no conflict of interest.

## Appendix A

**Table A1.** Means for lexical and hesitation marker front and central/back vowels.

|  | No of Items | Mean F1 | Mean F2 |
|---|---|---|---|
| **Lex Front** | 69 | 478.9 | 2053.8 |
| **HM Front** | 40 | 615.1 | 1861.9 |
| **Lex Central** | 78 | 466.3 | 1507.6 |
| **HM Central** | 185 | 616.8 | 1399.8 |

**Table A2.** Two-sample t-tests for lexical and HM vowels.

|  | *t-Value* | *df* | *P* | *Var* |
|---|---|---|---|---|
| **Front F1** | −6.6039 | 107 | <0.0001 | Equal |
| **Front F2** | 4.3417 | 106.08 | <0.0001 | Unequal |
| **Central F1** | −12.758 | 180.62 | <0.0001 | Unequal |
| **Central F2** | 3.8654 | 114.74 | 0.0001843 | Unequal |

## Notes

1    In what follows, the terms 'typological classification' or 'typological grouping' are used freely to stress the fact that the present approach does not involve genetic subgrouping; establishing 'typological regions' in Xiang here refers to defining regions by collections of shared features, rather than by establishing descent from a common parent language through a unique linguistic innovation. The former is essentially the traditional approach to Xiang classification (see e.g., Yuan 1960; Chen and Bao 2007), where regions are defined by bundles of shared features, whereas the latter is more recent (see e.g., Coblin 2011), although the two approaches are sometimes confused in Sinitic taxonomical studies.

2    For the purposes of this study, these subgroupings are treated as such, although some of the authors of the classificatory schemes may disagree.

3    Henceforth, just 'HMs' or 'hesitation markers' will be used to refer *only* to non-lexical HMs.

4    One may wonder if it is realistic to consider Cantonese representative of the form of hesitation markers in the Southern group, given that this group is quite diverse. However, in terms of the mid-vowel contrasts considered here, southern Sinitic varieties behave similarly; see Figure 2. Therefore, any other southern Sinitic variety of those mentioned in the discussion of Figure 2 could just as easily serve as a representative.

5    Shi et al. (2015) mention that Mandarin has a mid-front vowel approaching [ɛ] 'used only as an interjection' but have no instances of it in their corpus and do not provide an example.

6    The naming convention is to take the first syllable from the name of two dialects considered representative of the grouping; Changsha and Yiyang for Changyi, Loudi and Shaoyang for Loushao, and Jishou and Xupu for Jixu.

7    Map from: https://commons.wikimedia.org/wiki/File:China_blank_map_grey.svg (accessed on 21 November 2021).

8    This should not be interpreted as a claim that most northern dialects do not have a mid-front vowel *anywhere* in their inventories, or that southern dialects do not have mid-central vowels at all, only that they do not possess them contrastively in open CV syllables. For instance, some dialects may have one of these qualities obligatorily accompanied by an offglide or a final nasal,

　　and others may reduce vowels to schwa in fast speech; however, the current focus is on vowel qualities that have no such contextual restrictions.

9　　Each pair of brackets (separated by a comma if occupying the same square) represents a contrastive vowel quality in the dialect in question; if two qualities are within the same pair and are separated by a tilde <~>, this means the two qualities are in free variation.

10　　Map outline from: https://commons.wikimedia.org/wiki/File:Map_Hunan_adm.png (accessed on 21 November 2021).

11　　Strictly speaking, any lexical item can be lengthened to fill a pause, so any item from the lexical vocabulary might in theory serve an HM function. The examples provided here are predicted to occur most often as they are often used to introduce speech; neither list is exhaustive.

12　　Figure 6, Figure 7 and all statistical analysis conducted using R (R Core Team 2022).

13　　Standard deviation can be observed in Figure 6 as the dark gray shaded areas above and below the contour lines.

14　　All figures from this point created with Norm (Thomas and Kendall 2007).

15　　Possible exceptions to this generalization are PTH syllables such as [jɛ˦] 'leaf'; however, Duanmu (2007) points out that this vowel must be preceded by a palatal and categorizes the mid-front vowel as an allophone of /ɤ/.

16　　NE = 'northeastern', SW = 'southwestern', TR = 'transitional', HT = 'heritage'.

17　　By extension, the traditional Old-New classification (Yuan 1960) likewise cannot be supported by HM vowel data, even though this was not overtly explored in the study. This should follow from the high degree of similarity between Chen and Bao's (2007) classification and the traditional classification.

18　　An in-depth acoustic analysis, focusing solely on Mandarin HM vowels (restricted by location, e.g., Beijing), is necessary before the exact number can be determined. The observations of the two cited studies are too broad, as the analyzed regions in China are not restricted geographically, subsumed under the 'Mandarin' heading; however, even here, the number of vowel qualities seems more restricted than in the comparatively smaller Xiang-speaking region.

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
