# Peer review of "Vowel Quality in Xiang Non-Lexical Hesitation Markers: New Forms of Typological Evidence?"

_languages, doi:10.3390/languages8010073_

Round 1

Reviewer 1 Report

Please see the attached review report.

Reviewer 2 Report

The present piece of study has a solid contextualisation and foregrounding of research. It conducts a well-controlled testing against the data and assumptions from previous studies in historical-comparative linguistics and areal typology of Sinitic languages. The work is innovative and original in the sense that it incorporates a quantitative nature of phonetics into the typological investigation of Sinitic languages, a method which has not been practiced sufficiently hitherto in Chinese linguistics. It is a valuable attempt as phonetics, like pragmatics, is a more spontaneous domain of language, which can reveal the underlying structural system, especially in the process of changing.

The present study is relevant for the ongoing discussion in the field of diachronic and areal typology of Sinitic, showing a good example of conducting micro-areal studies to supplement and confirm macro-areal tendencies hypotheticised in previous studies. Furthermore, the research deals with and places Chinese linguistics in a position where it can participate and contribute to the discussion at a general level of linguistic typology as well as pragmatic phonetics.

The structure of the piece is clear and precise. Central references in the relevant fields are well selected and logically cited. Explanation of data and methods are informative with very detailed sociolinguistic description of the studied population and potential significant variables. All factors influencing any deviation and unexpected results is well elaborated on with convincing arguments and evidence.

Regarding readability, the contextualisation of the study can be reinforced by justifying even more firmly choices made in the research design. For instance, an argument for choosing non-lexical over lexical hesitation markers from 301 to 312 is very well stated and essential for the contextualisation of the present piece, so it may deserve to be raised immediately in the Introduction to strengthen the theoretical contribution and relevance beyond Chinese linguistics. As it is now, the contribution to Chinese linguistics seems to overshadow slightly that of pragmatic phonetics, even though the data and analysis equally well target at answering questions in the both fields. A similar argumentational reinforcement could be considered in other parts of the piece too where it can convincingly rule out unemployed alternative ways of conducting this research.

Author Response

The reviewer suggests a few areas of improvement, mostly dealing with the contextualization of the piece, particularly in the introduction. More specifically, the reviewer suggests highlighting the discussion on the lexical/non-lexical distinction, giving it more attention in the introduction, and suggests outlining ways in which the piece contributes to phonetic research (rather than Chinese linguistics). 

In incorporating these suggestions, the author has moved a discussion of this distinction to the introduction (lines 60-2), and included a few more lines discussing applicability of the research to acoustics of naturalistic speech (lines 90-3). The original discussion of the non-lexical/lexical distinction has also been expanded on lines 266-80 with additional examples/discussion. Additional contextualization of the approach is also provided nearer the conclusion, on lines 807-15. Finally, additional comments on contributions to phonetics were added on lines 861-7 to the conclusion.

Reviewer 3 Report

The article studies the vocalic qualities of HMs in samples of Xiang dialects. The two goals of the study, claimed by the author(s), are to explore the variations found in the vocalic qualities of the HMs across regions (sub-dialects) of Xiang and to determine whether such variations found in HMs can serve as evidence for the typological status of the dialect. However, the two goals are not reached with the research method, data collection, preparation, and argumentation of the study. 

The main concerns:

1)         The author(s) admit that there are considerable variations in HMs within the same speaker, between speakers, between social and regional dialects, etc. Yet this paper did not choose the proper methods that can tackle such huge variations. Instead, the research data was processed and presented as mostly means/averages/ranges, which don’t reflect the variabilities very well. 

2)         The conclusion of the paper states that the non-clear-cut fashion displayed in the vocalic qualities of the HMs in Xiang does not support the existing or the novel divisions of the dialect. One possibility, as the paper argues, is that these divisions are indeed inadequate. But due to the nature of HMs, the confounding factors, and the considerable variabilities in the data, we could also conclude that the HMs are simply not the best candidates to test these ways of divisions. In this case, the conclusion invalidates the paper itself. 

Minor concerns:

3)         The title of the paper can be more precise and concise. The paper does not involve “phonological patterns.” It is mainly about the vocalic qualities of the HMs.

4)         The language of the paper needs to be clarified and coherent. Many referential expressions, e.g., “this” and “here”, are used without more explicit context, which leads to logical confusion. 

5)         The word “typology” seems to be used interchangeably with “division,” which is not uncommon among Chinese dialectologists. However, typology in linguistics has a lot more than just “divisions” (of dialects). Using the term “typology” requires some clarification and justification.

Reviewer 4 Report

The reviewed article presents a thorough study of hesitation markers (HMs) in the Xiang variety of Chinese. The aim is to evaluate whether or not the vowel quality of HMs can contribute to internal typological taxonomy of Xiang. Although the results of the study do not provide a clear answer to the question of Xiang distribution, the article presents a new approach to the classification of dialects and can be a helpful addition to general studies on the typology of Chinese dialects.

The manuscript is clear and well-structured. The hypothesis is stated at the beginning and the research has been conducted applicably. The Author explains the need to study HMs as a neglected problem, and also as a possible tool in determining dialect boundaries. Although quite many publications on Xiang have been published so far, many of them are not very fresh (Yuan 1960 and later reeditions; Bao & Li 1985, etc.), and the classification and typology of Xiang certainly deserves further attention.

The cited references include most available studies on Xiang, although there are not many recent publications. That is probably due to the fact that not much research on the topic has been carried out lately. However, it would be advised to include not only the first, but also the second edition of The language atlas of China (中国社会科学院语言研究所,中国社会科学院民族学与人类学研究所,香港城市大学语言资讯科学院研究中心 2012. 中国语言地图集(第二版):汉语方言卷. 北京:商务印书馆.). The second edition holds a newer, much more detailed map of Xiang (map B-19) than the one found in 1988. There is also an article by Bao Houxing (pp.134-140) , which is a description of Xiang. The map corresponds with the one presented in the article as Map 1, but is dedicated only to Xiang, which makes it more detailed. Other works by Bao Houxing have been cited in the reviewed article, but possibly some additional information can be found in the Atlas.

The study which is the basis for the research, has been performed properly. The description of the methods of the experiment is precise. All the possible technical limitations or deficiencies have been mentioned by the Author and taken into consideration in the analysis of the results. The idea to record speakers of Xiang narrating a story is well justified as a possible source of HMs. The HMs provided by the task are an adequate basis for the research.

The tables, maps and diagrams are legible and explained comprehensively and are a proper illustration for the discussed topic. They also present the results of the experiment in an understandable manner. The conclusions of the article correspond with the hypothesis and the presented arguments. The Author is aware of the limitations of the research and that the problem of HMs, not only in Xiang, requires further study.

Minor editing in order to avoid repetitions could be required; e.g. in lines 366-369 the information about the data collected for the research is provided twice. This could be done if, for example, an allusion to the text above would be made. Also the statement of the genetic tenability of the Xiang by Coblin and Zhou & You is mentioned twice at almost the same place in the text (lines 132-138).

The level of English is appropriate. Only small linguistic errors, repetitions etc., which could require editing, can be found. One sentence is visibly unclear and needs editing (line 258: “includes items which have do not have other lexical functions other than indicating pauses”). If the language of the article is English, maybe pinyin should be added to Map 4?

The article provides a look on a very interesting feature of the language, the hesitation markers, which have not been widely researched, and as such presents an original perspective at Chinese dialects. The HMs can be a material for further study on their own, as a reflection of the phonetic inventory of a language/dialect.

As a side note, the problem of the wide-spread influence of Putonghua is made visible in the study. If the impact of the national standard is observable at the level of HMs, it must be present in most or all linguistics areas of dialects. This conclusion may encourage researchers to gather more data about the dialects and to try to protect them from endangerment.

The presented article, as an investigation on the question of typology of Xiang, and also as a proof of the diversity of Sinitic languages, will suit the special issue of Languages: Typology of Chinese Languages: One Name, Many Languages.

Author Response

The reviewer points out that the sources could benefit from some updating; it is true that many of the sources dealing with typological classification are somewhat outdated. The most recent publication on Xiang classification is Bao (2017), which turns out to be a collection of the author's writings, rather than a novel approach. I very much appreciate the reviewer making me aware of the updated 2012 version of the Language Atlas of China; a discussion of this version has been incorporated into the piece around lines 195-9, following the reviewer's suggestion. It will be noticed that the classificatory scheme is essentially the same as that found in Chen & Bao (2007) and in Bao (2017), with five distinct regions distinguished by traditional phonological criteria and novel lexical criteria.   

Minor Issues

The repetitions ('In total, 47 narratives from 16 dialect areas...' and the double reference to the work in Coblin and Zhou & You in the same paragraph) around lines 366-9 and 132-8 were removed.

The typo on line 258 has been corrected.

Pinyin has been added to Map 4.

Round 2

Reviewer 3 Report

The author took the reviewers' suggestions fixed the minor issues. 

But the main concerns that the reviewer raised persists.